# Eosinophilic inflammation: An Appealing Target for Pharmacologic Treatments in Severe Asthma

**DOI:** 10.3390/biomedicines10092181

**Published:** 2022-09-03

**Authors:** Alessandro Vatrella, Angelantonio Maglio, Corrado Pelaia, Luigi Ciampo, Giulia Pelaia, Carolina Vitale

**Affiliations:** 1Department of Medicine, Surgery and Dentistry “Scuola Medica Salernitana”, University of Salerno, 84100 Salerno, Italy; 2Department of Health Sciences, University “Magna Græcia” of Catanzaro, 88100 Catanzaro, Italy

**Keywords:** eosinophil, severe asthma, biologic drugs, type 2 inflammation

## Abstract

Severe asthma is characterized by different endotypes driven by complex pathologic mechanisms. In most patients with both allergic and non-allergic asthma, predominant eosinophilic airway inflammation is present. Given the central role of eosinophilic inflammation in the pathophysiology of most cases of severe asthma and considering that severe eosinophilic asthmatic patients respond partially or poorly to corticosteroids, in recent years, research has focused on the development of targeted anti-eosinophil biological therapies; this review will focus on the unique and particular biology of the eosinophil, as well as on the current knowledge about the pathobiology of eosinophilic inflammation in asthmatic airways. Finally, current and prospective anti-eosinophil therapeutic strategies will be discussed, examining the reason why eosinophilic inflammation represents an appealing target for the pharmacological treatment of patients with severe asthma.

## 1. Introduction

Asthma is a widespread chronic airway disease characterized by variable expiratory airflow limitation and bronchial hyperresponsiveness sustained by inflammation and remodeling of the airways [1].

Asthma is a complex and heterogeneous disease with considerable variations between different individuals and different possibilities of clinical presentation in terms of severity, age of onset and allergic condition [2]; these clinical characteristics are the expression of different inflammatory endotypes, which are the result of complex cellular and molecular pathobiological pathways including eosinophilic, neutrophilic, mixed and paucigranulocytic cellular patterns [3,4].

These endotypes are also involved in the pathogenesis of severe asthma [5], a clinical condition in which patient control requires treatment with medium or high dosages of inhaled corticosteroids (ICS)/long-acting β2-adrenergic agonists (LABA) combinations eventually supplemented with other drugs such as long-acting muscarinic antagonists (LAMA), leukotriene modifiers, oral corticosteroids (OCS) and/or targeted biological molecules [6,7].

The diagnosis of severe asthma is often not easy and requires an accurate assessment of the patient’s clinical and functional characteristics, as well as the levels of biomarkers such as peripheral eosinophilia or FENO (fractional exhaled nitric oxide) and the possible presence of comorbidities. Severe uncontrolled asthma also tends to be refractory to treatment, despite adherence to therapy and correct inhalation technique [6]. Exacerbations also play an important role in the natural history of the disease. Exacerbations are defined as episodes of worsening symptoms, associated with airflow obstruction and requiring patients to intensify treatment or go to the hospital, and more than 50% of patients with severe asthma have been shown to have frequent exacerbations [8,9], to the point of that the studies were completed in order to highlight if there were particular characteristics capable of defining the phenotype of frequent exacerbation. Interestingly, among these studies, a history of smoking, high short-acting β2-adrenergic agonists (SABA) use, the presence of sinusitis, and a lower age of onset of asthma were identified as risk factors for developing a frequent flare-up phenotype in the asthma cohort. Serious European U-BIOPRED [10].

Clinical, functional, and biological characteristics have been identified to cope with the difficulties in diagnosing and managing patients correctly. Clinically, these patients report poor symptom control despite maximal therapy and may present with upper airway comorbidities such as chronic rhinosinusitis with nasal polyposis [11]. Functionally, persistent airflow limitation with air trapping is common in these patients. The search for biomarkers such as IgE and the presence of positive allergy tests can be useful in distinguishing allergic etiology. Furthermore, the serum dosage of eosinophils is particularly useful in clinical practice [12].

Indeed, relevant eosinophilic airway inflammation is present in many patients with severe asthma regardless of their allergic condition. In fact, the International Severe Asthma Registry (ISAR) showed that eosinophilic asthma could be present in the majority (over 80%) of subjects with the most severe clinical phenotypes [13,14].

The development of airway eosinophilia is dependent on pathological networks involving close interactions between innate as well as adaptive immunity occurring in type 2 (T2) asthma following the involvement of group 2 innate lymphoid cells (ILC2) and T helper 2 (Th2) lymphocytes; these cells produce several cytokines including interleukin-5 (IL-5), -4 (IL-4) and -13 (IL-13). IL-5 represents the main regulator of the biology of eosinophils, including their differentiation, maturation, survival and activation [15].

Considering the pivotal role of eosinophilic inflammation in the pathophysiology of severe T2 asthma and given that patients with this type of asthma respond partially or poorly to corticosteroids (CS), in recent years research has focused on the development of targeted anti-eosinophil biological therapies. Pharmacological strategies against IL-5 or its receptor have fulfilled a crucial unmet need for the treatment of CS-dependent severe eosinophilic asthmatic patients who are greatly affected by side effects of oral corticosteroid (OCS) maintenance therapy.

In this article we will examine the current understanding about the pathobiology of eosinophilic inflammation, discussing the reason why this pathologic process represents an appealing target for the pharmacological treatment of severe eosinophilic asthma.

## 2. Eosinophil Structure and Biology

Eosinophils are granulocytes with unique biology. The fact that these cells have been largely preserved during evolution strongly suggests that they play relevant physiological functions. Eosinophils have traditionally been classified as effector cells with prevalent cytotoxic activity, although recent evidence indicates that these cells may play a role in a wide range of homeostatic and regulatory functions [16].

In homeostatic conditions, 0 to 500 eosinophils per microliter can be found in the blood, with a limited lifespan of 8–18 h in circulation. In inflammatory condition, eosinophils can be triggered and subsequently release their contents as a result of the action of various agents, including cytokines, lipid mediators and proinflammatory molecules.

### 2.1. Eosinophil Structure

Eosinophils are polymorphonuclear leukocytes, usually measuring 10–16 μm in diameter, with a segmented bilobed nucleus. Characteristic of this cell is the presence of a large number of molecules with pleiotropic functions, such as cationic granule proteins, chemokines, cytokines, growth factors, immunomodulatory molecules, lipid mediators, mainly accumulated within the intracellular compartment (Figure 1). Eosinophils also have a large array of transmembrane proteins (integrins) and surface receptors which mediate the interaction with the micro-environment and allow the response to multiple stimuli (Figure 1) [17].

#### 2.1.1. Eosinophil Granules

Eosinophil cytoplasm is packed with different types of granules (Figure 1). The two major types of large granules present within mature human eosinophils are specific granules and immature specific granules. The specific granules, also called “secondary granules”, consist of a dense crystalline nucleus surrounded by a membrane, a unique morphology found only in eosinophils. [18]; these granules contain a large variety of mediators, including basic proteins, cytokines, chemokines, growth factors and enzymes, which are able to produce tissue inflammation and damage. The main represented specific granule substances are eosinophil cationic protein (ECP), major basic proteins (MBP-1 and MBP-2), eosinophil peroxidase (EPX) and eosinophil-derived neurotoxin (EDN) [19]. Specific immature granules, also called “primary granules” are tendentially smaller than specific granules and are the principal location of Charcot-Leyden crystal (CLC) protein (a member of the carbohydrate-binding family of galectin-10).

In eosinophils have also been identified a third intracellular compartment, the lipid bodies, is specifically committed to the production of eicosanoid mediators of inflammation. 

Eosinophil sombrero vesicles (EoSVs) are not granules, but distinct tubular vesicles that tend to curl into a hoop-like morphology, giving rise to the term. EoSVs derive from specific granules and travel to the cell membrane to discharge their contents to the extracellular domain.

#### 2.1.2. Eosinophil Surface Receptors

Eosinophils display a vast array of receptors and surface molecules, which allow them to integrate with the innate and adaptive branches of the immune system involved in inflammatory responses and homeostasis. While many are selectively expressed on eosinophils such as interleukin-5Rα, CC-chemokine receptor 3 (CCR3), sialic acid-binding immunoglobulin-like lectin 8 (Siglec-8), the epidermal growth factor-like module containing mucin-like hormone receptor 1 (EMR1) appears completely unique to the eosinophil. The wide range of receptors present on eosinophils makes theme very versatile cells, with the ability to react to the stimulus, co-stimulate cells in antigen presentation and migrate to tissues in both physiological and pathological conditions [17]. 

Cytokine Receptors

Eosinophils display receptors for IL-3, IL-5, and granulocyte–macrophage colony-stimulating factor (GM-CSF), the three main cytokines involved in differentiation and maturation of these cells. The heterodimeric receptor for IL-5 is likely to be the most important cytokine receptor expressed by eosinophils, since IL-5 plays a fundamental role in all stages of eosinophil biology. The alpha-subunit, IL-5Rα, is specific to IL-5, while the beta-subunit is shared with the receptors for IL-3 and GM-CSF. Eosinophils also possess specific receptors for various other cytokines and growth factors, including IL-4, IL-13, IL-33, thymic stromal lymphopoietin (TSLP), and transforming growth factor-β (TGF-β) [18].

Adhesion Receptors

Eosinophils express various types of membrane adhesion receptors, primarily integrins and selectins, which are up regulated by a wide range of pro-inflammatory cytokines and chemokines. 

Integrin molecules are trans-membrane glycoproteins made up of an α and a β chain that includes the very late antigen-4 (VLA-4, CD49d/CD29) and the complement receptor CR3 (CD11b/CD18), also known as macrophage-1 (Mac-1) antigen.

Selectins are surface glycoproteins belonging to three groups (E-, L-, and P-selectin), in particular, L-selectin (CD62L) and P-selectin glycoprotein ligand-1 (CD162) are constitutively and highly expressed on circulating eosinophils [20].

Adhesion molecules act in a coordinated way, allowing eosinophils to roll and adhere to endothelia, thus facilitating their migration and accumulation at the sites of inflammation.

Chemoattractant Receptors

Eosinophils express on their surface various seven-transmembrane spanning G protein-coupled receptors for chemokines. Among them, CCR3 is an important, highly expressed receptor that binds to all three subtypes of eotaxin (a selective eosinophilic chemo-attractant) and to other chemokines, including the monocyte-3 chemoattractive protein (MCP-3) and MCP-4. The relevant role of CCR3 in asthma pathology is also supported by the evidence that the airways of patients with asthma contain more cells expressing mRNA for CCR3 and its ligands than non-asthmatic controls [21]. CCR1 is another key chemokine receptor on the surface of eosinophils, activated by chemoattractant cytokine ligand-3 (CCL-3) and CCL-5 (also known as RANTES: regulated on activation, normal T cell expressed and secreted).

Fc Receptors

The eosinophil displays various immunoglobulin (Ig) receptors and related family members involved in functional activities in which eosinophils are involved, including antibody-mediated cellular cytotoxicity (ADCC) for helminths and other immunomodulatory functions and pathological activities in diseases associated with eosinophilia. Fc receptors for IgA, IgD, IgE, IgG and IgM, localized on the membrane of eosinophils, promote interaction with the adaptive immune system.

The high-affinity Fc-epsilon R1-alpha (FcεR1) binding IgE is usually expressed in very small quantities in a trimeric form (without a β chain) and seems to have no role in eosinophil activation [22]. In contrast, cross-linking of FcαRI and FcγRII, with IgA and IgG, respectively, has been shown to induce eosinophil activation [23].

Major Histocompatibility Complex-II

Eosinophils express major histocompatibility complex class II (MHC-II) and co-stimulatory molecules such as CD80 and CD86, necessary for T-cell activation and proliferation. Lung eosinophils of asthmatic patients undergoing allergen challenges express higher levels of HLA-DR (a subtype of the MHC-II molecule) than blood eosinophils [24]. 

Pattern Recognition Receptors (PRRs)

Pattern Recognition Receptors are membrane proteins expressed on the surface of eosinophils that are directly stimulated during host innate immune responses from pathogen-associated molecular patterns (PAMPs) and damage-associated molecular patterns (DAMPs); these PRRs promote the interactions of eosinophils with invading microorganisms (especially helminths) and with the surrounding microenvironment. Among a variety of homeostatic and anti-infective activities, PRRs regulate the immune response and tissue damage [25]. Toll-like receptors (TLRs) are one of the most represented subtypes of PRR expressed by eosinophils (as well as many other cell lines) on their surface and even on endosomes [26]. Other significant PRRs expressed by eosinophils also include proteinase-activated receptors PAR-1 and -2. The latter could play a relevant role in the activation of eosinophils in response to proteases released by aeroallergens such as dust mites, fungi, or pollen [18]; moreover, there are other types of PRRs with partially overlapping characteristics of the previous mentioned, like retinoic acid-inducible gene-I-like receptors, nucleotide-binding oligomerisation domain-like (NOD-like) receptors, and the receptor for advanced glycation end products (RAGE) [25].

Lipid Mediator Receptors

Eosinophils express specific receptors for lipid mediators such leukotrienes, prostaglandins, and platelet-activating factors involved in eosinophil chemotaxis and transmigration.

Siglec-8

Sialic acid-binding immunoglobulin-like lectin (Siglec)-8 is an inhibitory receptor selectively expressed on human eosinophils, but information about its function in asthma pathology is still limited. Siglec-8 gene expression in asthma sputum cells is associated with type 2-high profiles of asthma and recent observation that administration of an antibody targeting Siglec-8 can induce selective eosinophil apoptosis, suggesting that it could represent a potential therapeutic target for eosinophil-mediated disease [27].

Inhibitory Receptors

Other inhibitory receptors regulating the survival and the activation of eosinophils, include CD300a, killer activating receptors, potassium inwardly-rectifying channel, and FcgRIIb [24].

#### 2.1.3. Intracellular Receptors

The eosinophil has many intracellular receptors that regulate its function (such as some toll-like receptors and the glucocorticoid receptor). In the intracellular compartment of eosinophils, a splicing variant of the glucocorticoid receptor (GR-A) is present in large quantities [28]. GR-A is the pro-apoptotic isoform and is five times higher in eosinophils than in neutrophils, which is why eosinophils are much more susceptible than other cells to the therapeutic actions of glucocorticoids, such as apoptosis [29]. 

### 2.2. Eosinophil Biology

A series of sequential processes regulate the particular biology of eosinophils; these events occur in different compartments, from the bone marrow to the blood and peripheral tissues, in physiological or pathological conditions. All the different phases, from maturation to degranulation, are regulated by the interaction of the eosinophil with a series of molecules that include transcription factors, adhesion molecules and cytokines.

#### 2.2.1. Eosinophil Differentiation and Maturation

Eosinophils are generated and developed in the bone marrow from multipotent hematopoietic stem cells, which create a population of committed progenitors of the eosinophilic lineage (EoPs) that in turn are capable of further differentiating into mature eosinophils, their terminal form [30].

Human EoPs are characterized by the expression of surface receptors such as CD34, CD38, and mainly high-affinity α subunit of the IL-5 receptor (IL-5Rα, or CD125) [31]. Differentiation of eosinophils normally occurs in the bone marrow; however, eosinophils can also develop from CD34+ EoPs which are found outside the bone marrow, blood and particularly lung tissue [32,33]. Increased levels of EoPs been identified in peripheral blood of atopic subjects compared to non-atopic controls [34]. Likewise, an increase in the number of CD34+/IL-5Rα+ EoPs has been identified in the bronchial mucosa of asthmatics compared to non-asthmatic controls [30]; moreover, the demonstration that blood EoPs have a greater response in vitro to IL-5 in patients with severe eosinophilic asthma than in milder asthmatics suggests a possible clinically relevant role of in situ eosinophilopoiesis in severe eosinophilic asthma [35].

Under homeostatic conditions, in healthy subjects, eosinophilopoiesis is mainly regulated by multiple transcription factors including GATA-binding protein 1 (GATA-1), Purine Rich Box-1 (PU.1), and the CCAAT-enhancing binding protein (c/EBP) family [36]. GATA-1 is thought to have the most important role, as disruption of the GATA-1 gene in mice results in a strain completely devoid of eosinophils [37].

The development of mature eosinophils in blood and peripheral tissues also depends on the synergistic contribution of cytokines such as IL-5, IL-3 and GM-CSF [18,38,39].

Eosinophils are fully differentiated after 7 days of maturation in the bone marrow. Mature cells are subsequently released in peripheral circulation, with a lifespan up to 24 h [40,41].

#### 2.2.2. Eosinophil Migration and Activation

Under physiological conditions, the main migration site of eosinophils is the gastrointestinal tract from the stomach to the intestine. A minority of them also migrate to the lymph nodes, thymus, liver, spleen, uterus and mammary gland [41,42].

This recruitment in different tissues can take place as early as 8–12 h after entry into circulation, mainly through the binding of the CCR3 to different chemokines, such as CCL11 (eotaxin-1), CCL24 (eotaxin-2) and CCL26 (eotaxin-3) [42].

When within tissues, eosinophils can survive for up to 15 days, primarily exerting an immunomodulatory function, but also promoting tissue repair as well as antimicrobial and antifungal immunity [41,42]. In inflammatory conditions, eosinophils can infiltrate tissues where they are not normally found or are minimally present, such as large and small airways and esophagus [43], in consequence of a series of extremely well-established steps. In particular, during allergic inflammation and bronchial asthma, circulating eosinophils adhere to the vascular endothelium and roll along it before pouring into lung tissue. Initial contact with the endothelium depends on the binding of the eosinophil cell membrane P-selectin glycoprotein ligand-1 (PSGL-1) to the adhesion receptor P-selectin to the activated endothelium [44]. The binding of integrin very late antigen-4 (VLA-4) to the vascular cell adhesion molecule-1 (VCAM-1) promotes the activation and extravasation of eosinophils [44]. IL-13 causes increased eosinophilic expression of P-selectin and increased P-selectin-mediated adhesion to endothelial cells [18,45].

In patients with severe asthma EoPs can also migrate to airways, where they differentiate to mature cells in situ [35].

Under physiological conditions, activation, survival, and recruitment of eosinophils are largely driven by IL-5, a cytokine produced by type 2 helper T cells (Th2) that plays a prominent role in the regulation of eosinophils, EoPs, mast cells and type 2 innate lymphoid cells (ILC2). Indeed, IL-5 is a key regulator cytokine for eosinophils acting at multiple functional levels and time points during their lifespan. Epithelium-derived alarmins, including IL-33, IL-25, and thymic stromal lymphopoietin (TSLP) partly trigger the production of IL-5 [46,47,48,49].

In the context of eosinophilic asthma, the increase in eosinophils in the airways begins after exposure of the epithelium of the airways to various allergens or antigens, thus triggering the activation of an immunological cascade that directs eosinophils into the airways through the stimulation by Th2 cytokines and chemoattractants [50]. When helper T cells are activated by allergens, they switch to the Th2 phenotype and begin secreting IL-4, IL-5, and IL-13 [22,40].

IL-5 and RANTES are the most relevant inducers of eosinophil migration in the asthmatic lung [51]. The airway epithelium is also involved in the secretion and production of these Th2 cytokines through the production of IL-33, and IL-25, which are secreted after any type of epithelial insult [52,53,54]; these alarmins also activate ILC2s from the innate immune system, which also secrete and produce IL-5, IL-4, and IL-13 [55]; it is interesting to consider that the action of the epithelial alarmins IL-25, IL-33 and TSLP on eosinophilopoiesis is both indirect, since the secretion of IL-5 occurs by the ILC2, but it is also direct, since IL-33 can precede and promote IL-5 signaling in the eosinophil development process [56].

In addition the relevant role in promoting the proliferation, differentiation, and maturation of EoPs expressing IL-5Rα in the bone marrow, IL-5 is able to induce the release of eosinophils into the bloodstream, as well as the activation and survival of mature eosinophils at the tissue level, acting in combination with eotaxines [57]. 

The cytokines IL-3 and GM-CSF are also implicated in the activation and survival of tissue eosinophils through induction of Bcl-xL expression, but their action is less specific than that of IL-5 [16,58,59].

#### 2.2.3. Eosinophil Degranulation

Once activated, eosinophils migrate from the peripheral blood to the inflammation site, in which they modulate the inflammatory response through the release of granules and therefore their contents [44,51]. Different degranulation processes are able to release specific granule contents (Figure 2):

(a) conventional exocytosis, in which the fusion of the granule directly with the cell membrane determines the release of the content of the specific granule itself;

(b) compound exocytosis, another type of exocytosis in which intracellular fusion of granules occurs prior to interaction with the plasma membrane and extracellular release;

(c) piecemeal degranulation (PMD), the most common mechanism of eosinophil degranulation, in which vesicles (that can be round or tubular) are released from specific granules and move towards the cell membrane to unload their content into the extracellular space [60]. Tubular vesicles tend to fold into a peculiar morphology and are therefore called “sombrero vesicles” [18];

(d) Cytolysis, a rapid non-apoptotic cell death in which the formation of vacuoles within cells occurs, with rupture of the nuclear and plasma membrane, the subsequent release of nuclear DNA and deposition of specific intact granules in the extracellular space. In this way, eosinophils release eosinophil extracellular traps (EETs), which consist of DNA fibers from the cellular nucleus [61]. Activated eosinophils are also capable of rapidly releasing other substances in addition to granule proteins into the extracellular space, such as bactericidal traps obtained from the combination of mitochondrial DNA and granule proteins; this type of cell death, which is characteristic of eosinophils, is also known as EETosis (“eosinophil extracellular trap cell death”) [31,62].

Finally, eosinophils can release exosomes into the extracellular environment; it has been demonstrated that eosinophils of asthmatic patients release greater amounts of extracellular vesicles (EVs) than those released by the eosinophils of healthy subjects [63]. EVs are important mediators produced by cellular processes [64]; this evidence strengthens the hypothesis that eosinophilic exosomes can be considered independent functional units, since, even in the absence of the cell of origin, they seem to be able to feed eosinophilic inflammation; however, exosomes and microvesicles are not the same entity: exosomes are generated by the fusion of multivesicular bodies (MVBs) with the plasma membrane, and microvesicles are shed by the outward vesiculation of the plasma membrane [65,66]. The pathogenetic relevance of eosinophil-derived EVs makes them a potential diagnostic and phenotypic biomarker of asthma, in particular of severe eosinophilic asthma [67,68].

### 2.3. Eosinophil Heterogeneity

Eosinophils were previously thought to be terminally differentiated cells upon their release from the bone marrow into the bloodstream, instead latest evidence demonstrated that eosinophils are able to further differentiate and mature in peripheral tissues, resulting in sub-populations with distinct phenotypic and functional profile [69].

Previous evidence from Mesnil et al. showed in mouse models a large population of eosinophils with a distinctive ring-shaped nucleus, both in absence of inflammation and following the development of dust-induced airway allergy, demonstrating the existence of lung resident eosinophils (rEosinophils) [70]; these cells, exclusively found in the lung parenchyma, express the surface receptors CD62L and CD125, intermediate levels of Siglec-F and low levels of CD101. Interestingly, even if they present the IL-5 surface receptor and react to IL-5 in vitro, rEosinophils seemed not to depend on IL-5 for their development [71], whereas the development of allergen-induced inflammatory eosinophils (iEosinophils) is known to be dependent on the activity of IL-5 [70] In addition, rEosinophils appear to have a more regulatory gene profile than iEosinophils, and mice without rheosinophils showed increased Th2 responses to inhaled allergens. Indeed, in allergic conditions, a large number of iEosinophils showing a segmented nucleus is recruited [70]. iEosinophils express low levels of CD62L, intermediate levels of CD125 and elevated levels of Siglec-F and CD101 on their surface, and are mainly concentrated in peribronchial areas [71].

Unlike mouse models, humans appear to have different eosinophil subsets based on cell density, in particular, normodense and hypodense eosinophils have been identified [72].

Normodense eosinophils from healthy individuals generally sediment at a density of 1.082 g/mL [72], while an increased number of hypodense eosinophils with a reduced density of <1.082 g/mL have been found in blood, BAL and lung tissues of patients with severe eosinophilic asthma [73,74]; these hypodense eosinophils were originally interpreted as activated eosinophils [75], since it has been shown that when normodense eosinophils are stimulated with GM-CSF, IL-3 or IL-5 and in the presence of fibroblasts, they switch to hypodense [76].

Hypodense eosinophils have been considered a “true” functional and phenotypic subset [77,78], since they highly react to activating stimuli. Indeed, after activation, hypodense eosinophils show greater survival, adhesion, oxygen metabolism, superoxide production, and antibody-dependent cytotoxicity than normodense [75].

Taken together, these studies have been useful in demonstrating that human lung eosinophils can be heterogeneous, however more studies are required for a better understanding of these subpopulations [69].

If the hypothesis of the existence of different subsets of human eosinophils is correct, it can significantly affect the choice of treatment in patients with severe eosinophilic asthma, between the total eradication of the eosinophilic lineage and the control of their IL-5-dependent development program [79].

### 2.4. Overview of Eosinophil-Driven Pathological Conditions

Elevated peripheral blood and tissue eosinophil counts can be found in several conditions, mainly allergic, rheumatological, infectious and neoplastic pathologies. Other possible conditions associated with eosinophilia have also been described, such as drug hypersensitivity, hematological and autoimmune diseases. Activation of eosinophils and subsequent release of eosinophilic mediators (mainly cytokines and type 2 chemokines) are potent pro-inflammatory effectors [80] and a significant association has been established between eosinophilia and some systemic inflammatory diseases [81]. The involvement of eosinophilia in inflammatory pathologies of the gastrointestinal, vascular, locomotor, and in general of the various mucous surfaces of the organism is known in the literature, and of course, eosinophils play key role in the pathogenesis of both upper and lower airway inflammation, with a spectrum of pathologic manifestations ranging from allergic rhinitis with or without nasal polyposis to asthma, and even allergic bronchopulmonary aspergillosis [82].

## 3. Eosinophils in Pathobiology of Severe Asthma

The mechanisms that regulate the complex pathways involved in airway inflammation in asthma can be broadly dichotomized into two different endotypes: type 2 (T2) endotype, under the coordination of Th2 lymphocytes and ILC2, which produce IL-5, IL-4 and IL-13, and non-type 2 (non-T2) endotype in which eosinophilia is absent [3].

Eosinophils are important effectors of T2 severe asthma, acting downstream of the inflammatory cascade stimulated by molecular mediators produced following inflammatory stimuli both on an allergic basis and a non-allergic basis [15].

Differentiation, proliferation, activation, survival and degranulation of eosinophils are mainly regulated by IL-5 produced by eosinophils themselves, Th2 lymphocytes, mast cells, natural killer cells, and ILC2 [83,84]. The biological effects of IL-5 are mediated by the interaction of the cytokine with the IL-5 receptor (IL-5R) which includes a specific α subunit for IL-5 (IL-5Rα) and a non-specific βc chain that can bind IL-5, IL-3, and GM-CSF [85]. IL-5 binding to IL-5Rα determinates the formation of a binary IL-5Rα/c receptor complex, which drives the activation of an intricate signal transduction pathway, including the JAK2–STAT1/3/5 complex that leads to proliferation, and a heterogeneous group of kinases (Raf-1, MAPK, PI3K) which are responsible for numerous activities of IL-5 such as activation, degranulation, survival and inhibition of apoptosis of eosinophils [86,87]. High levels of IL-5 have been detected in serum of patients with severe asthma, in whom eosinophilopoiesis occurs not only in the bone marrow but also in the airways [86]. Furthermore, IL-5 promotes eosinophil migration into the airways [88] synergistically with eotaxins, which are powerful eosinophil chemoattractants [89]; moreover, in type-2 asthmatic patients, IL-5 stimulates eosinophils to interact with the extracellular matrix protein periostin, whose levels are upregulated when eosinophils infiltrate the airways [90]. Finally, as previously mentioned, IL-5 also represents a key signal for the degranulation of eosinophils [91]. The release of the eosinophil granule content favors the damage of airway epithelium and neural tissue, since granules contain cytotoxic proteins including eosinophil cationic protein, eosinophil peroxidase, major basic protein and eosinophil-derived neurotoxin [18,92].

Eosinophils play a key role in the development of the major pathophysiological changes associated with severe asthma, including mucus hypersecretion, tissue damage and remodeling of the airways and consequent hyperreactivity of the airways (Figure 3); these effects influence relevant clinical outcomes such as asthma severity and risk of exacerbations.

Several evidence suggest that eosinophils are also active in the airway remodeling process in asthma, which consists of structural changes in the airways following repeated damage and repair processes; these findings are in line with the hypothesis that eosinophils can also act as regulators of morphogenesis and tissue repair. In asthma, airway remodeling is characterized by smooth muscle hypertrophy, epithelial cell hyperplasia, goblet cell metaplasia, and reticular basement membrane (RBM) thickening by deposition of collagen, tenascin, and other proteins of the matrix, resulting in progressive loss of lung function [93]. In particular, RBM thickening appears to be more evident in eosinophilic asthma than in non-eosinophilic asthma [94]. The exact mechanisms underlying airway remodeling processes in eosinophilic asthma are not yet fully understood; however, elevated TGF-β1 levels in the airways of asthmatic patients suggest a possible role for eosinophils [95,96], which are one of the main sources of TGF-β1 and after their activation by IL-5 [15,96] are able to release large amounts of TGF-β1 at the site of inflammation [97]. Furthermore, eosinophils express other cytokines associated to airway remodeling, such as heparin-binding epidermal growth factor (HB-EGF), nerve growth factor (NGF), TGF-α and above all the Th2 cytokines IL-4 and IL-13 [98,99]. 

## 4. Targeting Eosinophils in Severe Asthma

Eosinophils are key biological targets for the treatment of severe eosinophilic asthma, as these cells play a major role in the pathobiology of asthma.

Corticosteroids and anti-IL-5 / IL5r biologics are currently the most effective anti-eosinophil therapies, although other drug strategies are currently being studied. Furthermore, given their involvement in type 2 inflammation and eosinophil recruitment, it is worth mentioning drugs that target cytokines such as IL-4, IL-13 and alarmins such as IL-25, IL-33 and TSLP; however, these drugs are not considered strictly “antieosinophils”, as they reduce eosinophil counts via indirect mechanisms.

### 4.1. Corticosteroids

Corticosteroids (CSs) are the most common and powerful anti-inflammatory agents used for the treatment of eosinophilic asthma [100,101]. The therapeutic effects of CSs depend on their binding to cytoplasmic glucocorticoid (GR) receptors, which are mainly represented by the functional isoform GRa, largely more expressed than the GRb variant, which is alternately spliced and dysfunctional [102,103]. As a result of the interaction with either inhaled or systemic corticosteroids, GRs dissociate from anchoring chaperone molecules such as heat-shock proteins, and the active drug-receptor complex then translocate to the nucleus of target cells [104]. Inside the nucleus, GRs bind as homodimers to specific DNA nucleotide sequences named glucocorticoid response elements (GREs) [105]. Following this binding, coactivating protein complexes or corepressors are recruited and their molecular interactions with GRs consequently lead to stimulation or inhibition of gene transcription [106]. Through these mechanisms corticosteroids switch on several genes which mainly encode anti-inflammatory proteins such as glucocorticoid-induced leucine zipper (GILZ), that suppresses the bioactivities of the key pro-inflammatory transcription factors activator protein-1 (AP-1) and nuclear factor-κB (NF-κB) [107]. Another important anti-inflammatory protein whose expression is up-regulated by corticosteroids is mitogen-activated protein kinase phosphatase-1 (MKP-1), which dephosphorylates and inactivates the pro-inflammatory p38 subgroup of mitogen-activated protein kinases (MAPKs) [108]. In addition to modulating gene expression at the GRE level, monomeric GRs can also directly bind NF-κB and AP-1, thus forming protein-protein complexes that repress the pro-inflammatory effects of these transcription factors by preventing their interactions with specific DNA consensus sequences [109]; this latter event may be promoted by corticosteroids also via GR-induced recruitment and activation of histone deacetylase-2 (HDAC2) [110], which results in deacetylation of core histones, nucleosomal DNA condensation, and the consequent inaccessibility of nuclear binding sites for pro-inflammatory transcription factors. Furthermore, corticosteroids can also act at a post-transcriptional level by up-regulating the expression of tristetraprolin, a zinc finger protein that destabilizes some cytokine mRNAs [111].

CSs possess a wide range of anti-inflammatory actions through the above molecular mechanisms, including suppression of eosinophil maturation, survival, proliferation, activation and chemotaxis, as well as induction of their apoptosis [112,113,114]. In particular, corticosteroids repress eosinophil differentiation and promote eosinophil apoptosis by blocking the production of key growth factors such as IL-3, IL-5 and GM-CSF [101,115]. Furthermore, corticosteroids are able to effectively inhibit the recruitment of eosinophils into the airways, mainly due to the CS-induced down-regulation of IL-13 synthesis, which leads to a marked decrease in IL-13-dependent production of the chemoattractive eotaxin [116]. CSs have the ability, through inhibition of the biosynthesis of eotaxin and RANTES (regulated upon activation, normal T cell expressed and secreted), to contribute to the removal of eosinophils from the airways, responsible for attenuation of antigen-induced bronchial eosinophilia [117]. In fact, in allergic asthmatics, CSs have demostrated to be able to significantly lower the number of eosinophils in induced sputum and this effect is associated with a strong reduction in the hyperreactivity of the airways to methacholine [118]. Furthermore, CSs reduce the attachment of eosinophils to bronchial epithelial cells and this further therapeutically useful action appears to result from CS-dependent inhibition of VCAM-1 expression in the airway [119]. CSs also decrease BAL levels of ECP, a highly cytotoxic molecule released by activated eosinophils [120]. 

All these pharmacological properties place CSs in inhaled formulation among the first-line drugs for the treatment of asthma, even in children [121]; however, despite the multiplicity of corticosteroid antieosinophilic actions above described, many patients with severe refractory eosinophilic asthma may exhibit various degrees of resistance to corticosteroids [122]. Many mechanisms can sustain corticosteroid resistance in asthma, including an exaggerated production of IL-5 and IL-13, an excessive expression of the dysfunctional GRb isoform, and an impairment of histone deacetylases [123,124]. A further cause of corticosteroid insensitivity is attributable to overactivation of p38 MAPK, which phosphorylates GRs thus compromising their nuclear translocation and DNA binding properties [125]. If inhaled or even oral corticosteroids are unable to provide satisfactory clinical control in patients with severe eosinophilic asthma, biological antieosinophilic therapies should be considered [116].

### 4.2. Anti-Eosinophil Biological Therapies

Based on the recommendations in step 5 of the Global Initiative for Asthma (GINA) document, currently available strategies for the treatment of severe asthma with direct anti-eosinophilic mechanism include inhibition of IL-5 and receptor antagonism of the receptor for IL-5 [126]; these pharmacological options are indicated exclusively in severe asthma, since there are currently insufficient data in the literature to prefer biological therapy to conventional treatment in less severe forms of disease, and furthermore the high cost of biologics limits their use in clinical practice in patients that respond to the treatments recommended in the lower GINA steps.

#### 4.2.1. IL-5 Inhibition

Mepolizumab is a humanized monoclonal IgG1/k antibody which binds to the α-chain of IL-5, thereby preventing its interaction with the α subunit of the IL-5 receptor (IL-5Rα) [127,128]. The efficacy of mepolizumab was initially demonstrated in some patients with severe eosinophilic asthma who experienced recurrent disease exacerbations [129,130]. In particular, in these subjects, mepolizumab significantly reduced asthma exacerbations and also lowered the number of eosinophils in the blood and sputum; these important findings were subsequently reinforced by the results of the phase 2b/3 DREAM (Dose Ranging Efficacy And safety with Mepolizumab) study, conducted in a larger population of patients with severe uncontrolled eosinophilic asthma [131]. In addition, the phase 3 trials MENSA (MEpolizumab as adjunctive therapy iN patients with Severe Asthma) and SIRIUS (SteroId ReductIon with mepolizUmab Study) showed a decrease in the number of severe asthma exacerbations, a higher quality of life, a better symptom control, and an increase in FEV1 in asthmatic subjects with the above features, treated with mepolizumab [132,133]. The SIRIUS study also showed that mepolizumab elicited a 50% decrement of OCS intake [133]. The additional phase 3b MUSCA trial corroborated the positive impact of mepolizumab on health-related quality of life [134]. Additionally, a post hoc meta-analysis of MENSA and MUSCA showed a mepolizumab-induced improved exercise tolerance and worked productivity in patients complaining of severe eosinophilic asthma [135]. After completing either MENSA or SIRIUS studies, many asthmatics were then enrolled in the COSMOS open-label, phase 3b extension trial, which confirmed a good long-term safety and efficacy profile of mepolizumab [136].

Recent real-world findings further validated the results of randomized controlled trials, even suggesting that in clinical practice mepolizumab can be more effective than previously shown, probably because of the higher numbers of blood eosinophils characterizing the patients involved in many real-life experiences [137,138,139]. In this context, mepolizumab super-responder patients, expressing very high levels of several biomarkers of type 2 inflammation, have been identified [140]. Real-world studies also indicate that mepolizumab shows similar efficacy in both atopic and non-atopic patients with severe eosinophilic asthma [141]; these observations support the use of mepolizumab as an effective biological treatment switch for eosinophilic allergic asthmatics unresponsive to omalizumab [142,143]. Additionally, mepolizumab has been shown to improve proximal and distal airway airflow along the respiratory tree of patients with severe eosinophilic asthma, as evidenced by significant increases in FEV1 and FEF25-75% [139].

Reslizumab is another humanized IgG4/k anti-IL-5 monoclonal antibody, which has been evaluated in several randomized controlled trials included in the BREATH program [144,145]. An initial phase 2 trial showed reduced blood and sputum eosinophil numbers, and also induced a transient FEV1 increment, after use of reslizumab [146]. An additional phase 2 study in patients with severe eosinophilic asthma showed that reslizumab was able to significantly improve FEV1 and to elicit a trend toward symptom control improvement, especially in patients having high blood eosinophil counts and comorbid nasal polyposis [147]. Subsequently, two phase 3 studies were conducted in severe asthmatics with more than 400 blood eosinophils/mL, showing a decrease in the annual rate of asthma exacerbations higher than 50%, with relevant improvements in symptom control and lung function, after reslizumab treatment [148]; these data were later confirmed in subjects complaining of late-onset eosinophilic asthma [149]. Similar to mepolizumab, reslizumab was also able to promote significant increases in both FEV1 and FEF25-75% [150]. Such positive therapeutic effects have been also confirmed by real-life experiences [151]. In consideration of the evidence from both randomized trials and real-life studies, reslizumab has been shown to be a biological drug with a good safety and tolerability profile [147]. Nevertheless, a pooled analysis of 6 trials including 1028 patients found that reslizumab caused 3 cases of anaphylaxis [152]. One of these occurred in a patient with a history of drug hypersensitivity, who experienced a reaction with dyspnea, chills, vomiting and hot flashes. In the second case, the anaphylactic reaction presented itself as shortness of breath, wheezing, inability to speak, swollen eyes, hot flashes, and desaturation. The third reslizumab-related anaphylactic reaction occurred in a patient with a history of drug allergy and hypersensitivity, with skin reactions, severe lower abdominal pain and severe burning and itching in the genital area [152].

#### 4.2.2. IL-5 Receptor Antagonism

The IL-5 receptor blocking strategy is effectively carried out by the humanized and afucosylated IgG1/k monoclonal antibody benralizumab, which through Fab fragments selectively binds to IL-5Rα, thus preventing its interaction with IL-5 [153]. Furthermore, the constant Fc region of benralizumab binds to the FcγRIIIa receptor located on natural killer (NK) cells, thereby triggering eosinophil apoptosis through antibody-dependent cell-mediated cytotoxicity (ADCC), a mechanism strongly enhanced by afucosylation [153]. Benralizumab, therefore, has a double antieosinophilic action, exerted both through the neutralization of IL-5 at the receptor level, and through a direct pro-apoptotic effect; this monoclonal antibody has been thoroughly tested across the extensive WINDWARD program which included the phase 3 SIROCCO and CALIMA trials, which demonstrated that benralizumab is able to significantly reduce the number of severe eosinophilic asthma exacerbations and improve symptom control and respiratory function [154,155]. The positive effect exerted by benralizumab on pulmonary function was also documented by the phase 3 BISE trial, which detected a quite relevant FEV1 increase in patients with eosinophilic asthma and a blood eosinophil count of at least 300 cells/mL [156]. Additionally, the phase 3b ANDHI study showed that benralizumab, in addition to reducing the number of asthma exacerbations and increasing both FEV1 and PEF, also improved health-related quality of life and symptom control [157]. ZONDA and PONENTE trials were very useful because they demonstrated the powerful OCS sparing effect of benralizumab in subjects with severe eosinophilic asthma [158,159]. Furthermore, the BORA Phase 3 extension study reported a high long-term drug safety and tolerability profile of benralizumab [160]. Both placebo-controlled trials and real-life observations suggest that benralizumab is an effective add-on biologic treatment for allergic and non-allergic people with severe eosinophilic asthma [161,162,163]. As a result, real-world studies have also shown that benralizumab can be successfully used as a biological switching therapy for allergic patients with severe eosinophilic asthma who are not fully responsive to omalizumab [68]. Combining the results of controlled and real-life studies of severe eosinophilic asthma, benralizumab has proven to clearly be a biologic drug that can deplete blood eosinophils, decrease asthma exacerbations and OCS consumption, as well as improve symptom control, airflow limitation and lung hyperinflation [162,163,164,165].

#### 4.2.3. Currently Available Biological Therapies Indirectly Targeting Eosinophils 

In addition to biological strategies aimed at inhibiting the proliferation and eliminating circulating eosinophils, there are currently other pharmacological agents capable of intercepting the pro-inflammatory and pro-remodeling activities of the eosinophil, thus acting as indirect anti-eosinophil drugs.

Dupilumab is a fully human IgG4 monoclonal antibody capable of suppressing the biological actions of IL-4 and IL-13 through selective binding to IL-4Rα, shared by these two cytokines for the activation of their receptor mechanisms [166]; this drug is effective in severe asthmatics with at least 150 eosinophils per microliter of blood and / or at least 25 parts per billion (ppb) of FeNO and has therapeutic effects including rapid and relevant improvements in asthma exacerbations, symptom control, airflow limitation, lung hyperinflation and OCS intake [167,168,169].

Tezepelumab is a fully human anti-TSLP monoclonal antibody recently approved by the FDA for the adjunctive maintenance therapy of severe asthma, without phenotype or biomarker limitation. In patients with moderate to severe asthma, tezepelumab can act independently of blood eosinophil counts, decreasing the rate of asthma exacerbation by 56% and improving symptom control, lung function and quality of life-related to health [170,171]. Very recent studies have shown that tezepelumab was also able to induce a marked decrease in the number of eosinophils in both BAL and bronchial biopsies obtained from adult asthmatics [172]. Finally, due to the ability of TSLP to favor the switch of naïve Th lymphocytes to a Th17 cell line, tezepelumab may have potential beneficial effects in the treatment of Th17 cell-driven non-type 2 asthma [173,174].

### 4.3. Experimental Anti-Eosinophil Therapies

The newly developed anti-eosinophil therapies include chemokine receptor antagonists, prostanoid receptor blockers and pro-apoptotic antibodies, as well as inhibitors of kinases and transcription factors, which are under ongoing evaluation in both animal and human studies.

#### 4.3.1. Chemokine Receptor Antagonists

Eosinophils express the CCR3 receptor, which is activated by the powerful eosinophil chemoattractant eotaxin [101]. Using an experimental mouse model of allergic asthma, a rat CCR3 anti-mouse monoclonal antibody was shown to effectively inhibit the ovalbumin-induced increase in eosinophil counts in both BALF and lung biopsies [175]. When administered before inhaled allergen challenge to patients with mild-to-moderate atopic asthma, the selective oral CCR3 antagonist AXP1275 elicited a not significant trend towards a decrease in antigen-dependent airway eosinophilia, associated with a significant improvement in bronchial hyperresponsiveness to methacholine [176]. Similar results were obtained utilizing GW766994, another oral CCR3 antagonist [177]. TPI ASM8 is a complex construct consisting of two modified phosphorothioate antisense oligonucleotides, including the CCR3 antagonist TOP005 and TOP004, which are directed against the common βc subunit of the receptors for IL-3, IL-5, and GM-CSF [178]. In comparison to placebo, TPI ASM8 significantly lowered by 46% allergen-induced airway eosinophilia in patients with mild allergic asthma, and also reduced early and late asthmatic responses [178]; moreover, TPI ASM8 dose-dependently suppressed the allergen challenge-induced increase in the airway concentration of eosinophil cationic protein [179].

#### 4.3.2. Antagonists of Prostaglandin D2 Receptor

Due to the widespread expression of the prostaglandin D2 receptor CRTH2 (chemoattractant receptor-homologous molecule expressed on Th2 cells) on several immune-inflammatory cells [180] including eosinophils, some oral CRTH2 antagonists have been tested in order to evaluate their potential therapeutic effects on eosinophilic inflammation in the airways [101]. When compared to placebo, the CRTH2 antagonist OC00049, subsequently identified as timapiprant, significantly decreased sputum eosinophil counts and increased FEV1 in patients with moderate persistent asthma [181,182,183,184]. In a preliminary phase 2 study carried out in patients with chronic eosinophilic asthma, the CRTH2 antagonist fevipiprant reduced sputum eosinophil numbers, and improved lung function and asthma-related quality of life [185]; however, in the subsequent LUSTER-1 and LUSTER-2 phase 3 trials fevipiprant was not able to guarantee in patients with severe asthma a significant reduction of disease exacerbations, which represented the primary study endpoint [186].

#### 4.3.3. Inducers of Eosinophil Apoptosis

Human eosinophils express on their cell membrane the inhibitory receptor named Siglec-8 [187]. The binding of the monoclonal antibody lirentelimab/AK002 to Siglec-8 triggered a mechanism of antibody-dependent cellular cytotoxicity leading to eosinophil apoptosis and to the consequent reduction of sputum eosinophils from asthmatic subjects [27]. Therefore, antibody-mediated induction of eosinophil apoptosis currently appears to be a promising therapeutic approach for the treatment of diseases characterized by eosinophilic inflammation, such as type 2 asthma [188].

#### 4.3.4. Inhibitors of Kinases and Transcription Factors 

Since proinflammatory type 2 cytokines exert their biological actions via activation of complex signaling modules mediated by JAK (Janus kinases)/STAT (signal transducers and activators of transcription) pathways [84,189], these molecules can be targeted by experimental therapies of eosinophilic asthma under current investigation. In particular, the JAK-3 specific inhibitors tofacitinib and CP-690550 markedly reduced the numbers of BAL eosinophils in murine models of allergen-induced lung inflammation [190,191]. 

The transcription factor GATA-3 is a suitable potential target for experimental therapies for type 2 asthma. Indeed, GATA-3 promotes the commitment of naïve Th0 cells towards the Th2 lineage thus up-regulating the production of type 2 cytokines, also including IL-5 [192,193]. In this regard, during the last decades, an interesting research strategy has evolved, leading to the development of GATA-3 DNAzymes, consisting of single-stranded catalytic DNA molecules which selectively cleave GATA-3 mRNA [194]. In atopic asthmatic patients, a placebo-controlled trial showed that the inhaled GATA-3-specific DNAzyme SB010 successfully attenuated allergen-induced sputum eosinophilia, and also decreased plasma levels of IL-5 [195].

## 5. Conclusions

Eosinophilic airway inflammation is a key “treatable trait” in patients with severe asthma. For patients exhibiting this trait who have an incomplete response to corticosteroids, novel biologic therapies have been developed and are available in clinical practice; these drugs have been shown to significantly improve symptom control, reduce exacerbation rates and oral glucocorticoid use in many but not in all severe asthmatic patients; it is not yet clear which patients will respond to which biologic agents. Thus far, it seems that anti-IL-5 is most efficacious in patients with high blood eosinophil counts; however, it is possible that eosinophils’ plasticity and adaptability and their pleiotropic responses to the most disparate stimuli are at the basis of the relative impossibility, at least with the currently available drugs, to definitively “cure” from severe asthma.

So, immediate future challenges should include determining which eosinophil-reducing treatment is more effective for patients with severe eosinophilic asthma. 

## Figures and Tables

**Figure 1 biomedicines-10-02181-f001:**
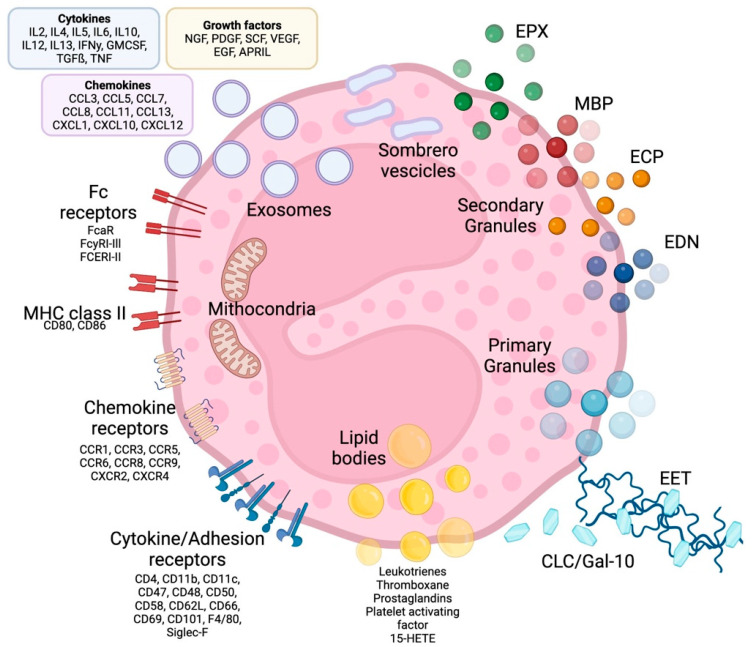
Eosinophil structure, granules, surface receptors, exosomes and EETs. Eosinophils are supplied by a large number of proteins, receptors and enzymes that allow them to interact with the microenvironment and express a number of receptors on their surface, including receptors for cytokines, chemokines and lipid mediators, which are involved in cell growth, survival, adhesion, migration and activation. In addition to receptors, adhesion molecules such as integrins are expressed on the cell surface, which allows eosinophils to migrate and react to several stimuli. The effects of eosinophils are largely achieved due to the content of their granules. Primary granules include Charcot-Leyden/galectin-10 protein, a characteristic eosinophilic protein implicated in asthma and parasitic infections, as well as a constituent part of so-called eosinophilic extracellular traps, whose other major constituents are nuclear or mitochondrial DNA strands. Specific or secondary granules contain four main cationic proteins: MBP, ECP, EPX and EDN. In addition, some of the content of the granules is released through particular vesicles called sombrero vesicles. Each of them has different effects, clarified in the text. Lipid bodies contain prostaglandins, thromboxane and leukotrienes, which participate in allergic inflammation, fibrosis and thrombosis. Finally, eosinophils are able to release exosomes that fuse with the cell membrane, which are involved in epithelial damage. CLC/Gal-10: Charcot-Leyden crystal proteins; ECP: eosinophil cationic protein; MBP: major basic proteins; EPX: eosinophil peroxidase; EDN: eosinophil-derived neurotoxin; MHC class II: Mayor histocompatibility complex-II; EET: eosinophilic extracellular traps. See the text for further explanation.

**Figure 2 biomedicines-10-02181-f002:**
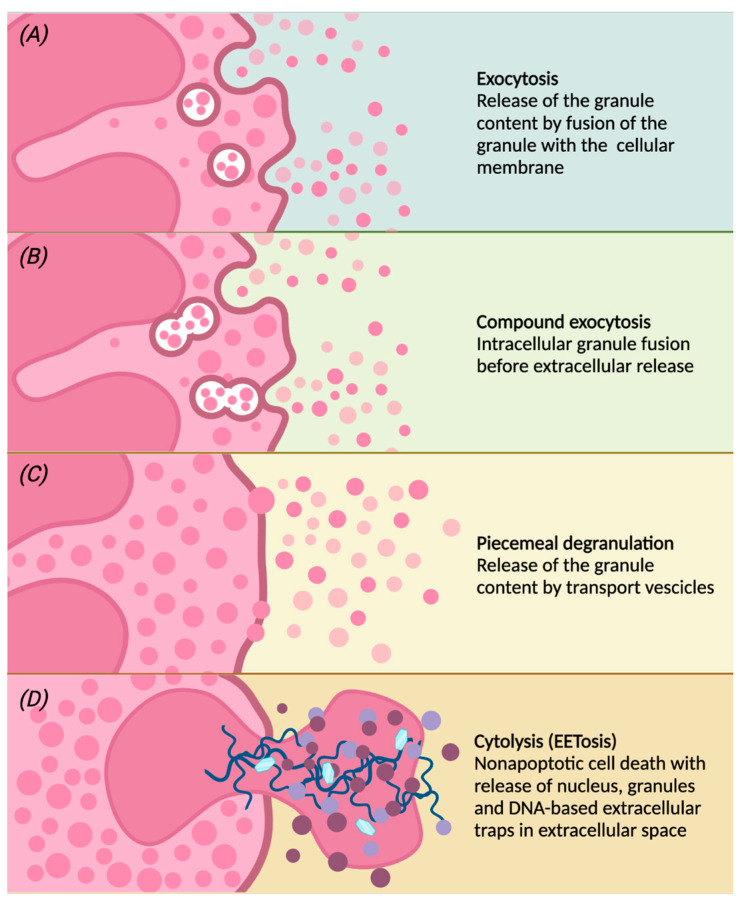
Following different stimulations, eosinophils can release the contents of the granules by classical exocytosis, compound exocytosis, piecemeal degranulation (PMD) or cytolysis. Conventional exocytosis consists of the release of granule content by fusion of the granule itself to the cellular membrane (panel **A**). Compound exocytosis is another type of exocytosis in which granules in which granules merge with each other before interacting with the cellular membrane (panel **B**). Piecemeal degranulation is the progressive and selective release of vesicles from specific granules and the unloading of their contents after the fusion with the cellular membrane (panel **C**). Cytolysis is a non-apoptotic form of cell death with rupture of the nuclear and plasma membrane, subsequent release of nuclear DNA and deposition of specific intact granules in the extracellular space. After cytolysis, there may be the release of eosinophilic extracellular traps (EETs), giving this peculiar form of cell death the characteristic name of EETosis (panel **D**).

**Figure 3 biomedicines-10-02181-f003:**
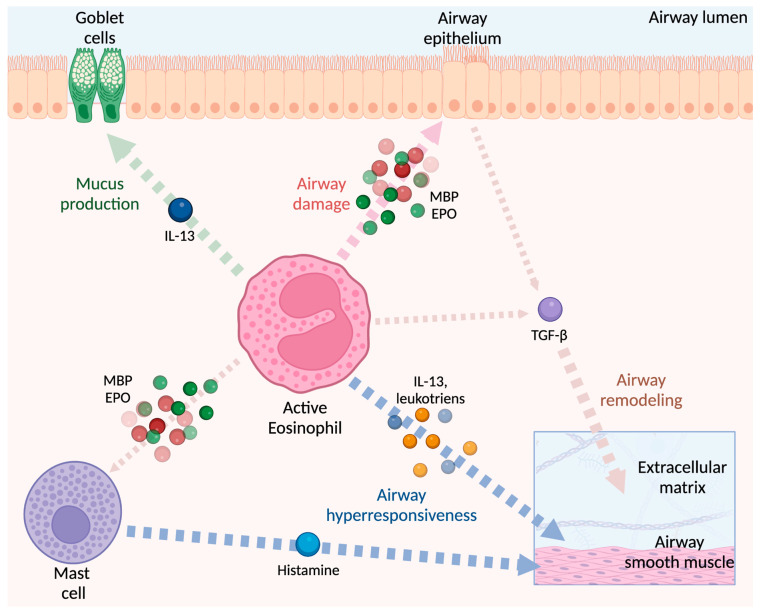
The centrality of the eosinophil in the pathophysiology of T2 asthma. The main effect can be summarized in four peculiar activities: airway damage, airway remodeling, airway hyperresponsiveness, and mucus production. When activated eosinophils reach the airways, they release specific granules whose contents have cytotoxic properties that can cause direct damage to the airways. The content of these granules, especially MBP and EPO, is also able to stimulate mast cells and basophils to release histamine, which contributes to bronchial hyperresponsiveness together with the direct action triggered by the release of IL-13 and leukotrienes by the eosinophil itself. IL-13 also increases mucus secretion by promoting the differentiation of goblet cells. Airway remodeling is associated with smooth muscle cell hyperplasia and fibroblast proliferation, which are promoted by TGF-β released both as a result of epithelial damage and as an exosomal content of eosinophils. TGF-β is also responsible for structural changes to the extracellular matrix, by increasing the production of collagen and glycosaminoglycans. IL-13: interleukin-13; MBP: mayor basic protein; EPO: eosinophil peroxidase; TGF-β: transforming growth factor-β.

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
