# Peer review of "Eosinophilic inflammation: An Appealing Target for Pharmacologic Treatments in Severe Asthma"

_biomedicines, 2022, doi:10.3390/biomedicines10092181_

Round 1

Reviewer 1 Report

The manuscript provides a detailed, well structured overview on eosinophils and eosinophilic inflammation and also provides some insight into severe asthma and the use of anti-eosinophil treatments. Overall it represents a useful, timely contribution to the relevant scientific literature.

Comment:

It would be useful to provide a hint on the wide scale of eosinophilic inflammations in the body and bring focus from the overall picture to the airways and asthma.  I also suggest a bit more detailed introduction of severe asthma and position eosinophilic airway inflammation in its pathomechanism. There is rapidly growing evidence on different pathways published by a huge collaborations like U-BIOPRED and SARP and some others  as well that continuously form our understanding of the disease.

My suggestion is the include some references pinpointing eosinophil involvement, changing endotypes, etc. Publications such as Hoda U et al Clin Transl Med 2022; provides important insigth to streoid treated, but high eosinophil asthma that is very relevant to deliniate the problem of liitied efficiency of steroids in severe asthma. There are expert consensus statement such as Severe eosinophilic asthma: A roadmap to consensus”, that are also worthy to add for better understanding.

In the list of anti-eosinophil treatment it would be good to see something on anti-IL4, anti IL-13s,  dupilumab and tezepelumab and also some details on fevipiprant and timapiprant.

Finally a brief paragraph on the suspected reasons why none of the anti-eosinophil treatments could erase severe asthma.

Minor comments:

line 30 text: „These clinical are…” lacks a word after clinical.

line 34 the word „sustain” is not appropriate in the given sentence.

line 555. Regarding anaphylaxis it would be useful to add some lines on t he potential mechanism (is there anything tod o with eosinophils?).

Author Response

We thank the reviewer for having positively evaluated our work and we are pleased to be able to increase the quality thanks to the suggestions he has provided.

We appreciate the reviewer's suggestions, and we provide our point by point responses below:

Q: It would be useful to provide a hint on the wide scale of eosinophilic inflammations in the body and bring focus from the overall picture to the airways and asthma.  

A: Following your hint, we added an apposite paragraph on lines 419-432.

Q: I also suggest a bit more detailed introduction of severe asthma and position eosinophilic airway inflammation in its pathomechanism. There is rapidly growing evidence on different pathways published by a huge collaborations like U-BIOPRED and SARP and some others  as well that continuously form our understanding of the disease. My suggestion is the include some references pinpointing eosinophil involvement, changing endotypes, etc. Publications such as Hoda U et al Clin Transl Med 2022; provides important insigth to streoid treated, but high eosinophil asthma that is very relevant to deliniate the problem of liitied efficiency of steroids in severe asthma. There are expert consensus statement such as "Severe eosinophilic asthma: A roadmap to consensus”, that are also worthy to add for better understanding.

A: We appreciate your advice and in the introduction we have deepened the ideas proposed, in particular in lines 40-63. We have also provided to cite the suggested references.

Q: In the list of anti-eosinophil treatment it would be good to see something on anti-IL4, anti IL-13s,  dupilumab and tezepelumab and also some details on fevipiprant and timapiprant.

A: We have added an apposite paragraph for Dupilumab and tezepelumab at lines 665-686 and we discussed about the use of timapiprant and fevipiprant on lines 712-725. Regarding both topics, we have also expanded the corresponding bibliography.

Q: Finally a brief paragraph on the suspected reasons why none of the anti-eosinophil treatments could erase severe asthma.

A: We briefly added a possible explanation as to why none of the anti-eosinophil treatments could clear the severe asthma in the lines 760-765.

Q: line 30 text: „These clinical are…” lacks a word after clinical.

A: corrected.

Q: line 34 the word „sustain” is not appropriate in the given sentence.

A: corrected.

Q: line 555. Regarding anaphylaxis it would be useful to add some lines on t he potential mechanism (is there anything tod o with eosinophils?).

A: We have added what required in the lines 625-631.

We hope that the reviewer finds the answers satisfactory and we take this opportunity to extend our best regards.

Reviewer 2 Report

This is an interesting review article on severe Asthma and Eosinophilic inflammation. The authors have done an excellent job summarizing the central role of eosinophilic inflammation in Asthma. They also discussed biological therapies that have been developed and in clinical practice. In my opinion, I would add to the article why anti-IL-5 is not suitable for patients who have moderate persistent Asthma. 

Author Response

We thank the reviewer for the opinion expressed regarding our work and especially for the valuable advice on how to improve it. 

In relation to the suggestion provided, we have added to the article a small introduction to anti-eosinophil drug treatment in which we explain why anti-IL-5 is currently not suitable for patients with less severe forms of asthma (lines 565-572).

We hope that the reviewer finds the answer satisfactory and we take this opportunity to extend our best regards.